# Early life adversity, contact with children's social care services and educational outcomes at age 16 years: UK birth cohort study with linkage to national administrative records

Alison Teyhan ![ORCID],[1] Andy Boyd,[1] Dinithi Wijedasa,[2] John Macleod[1]

¹Bristol Medical School, Population Health Sciences, University of Bristol, Bristol, UK
²School for Policy Studies, University of Bristol, Bristol, UK

**Correspondence to**
Dr Alison Teyhan;
alison.teyhan@bristol.ac.uk

## ABSTRACT

**Objectives** To use record linkage of birth cohort and administrative data to study educational outcomes of children who are looked-after (in public care) and in need (social services involvement), and examine the role of early life factors.

**Setting, design** Prospective observational study of children from the Avon Longitudinal Study of Parents and Children (ALSPAC), which recruited pregnant women in and around Bristol, UK in the early 1990s. ALSPAC was linked to the annual Children Looked-After (CLA) Data Return and Children In Need (CIN) Census. Educational outcomes at 16 years were obtained through linkage to the National Pupil Database (NPD). These included passing 5+ good GCSEs (grades A*-C, including English and Maths). Covariates included early life adversity and social position.

**Participants** 12 868 ALSPAC participants were linked to the NPD. The sample for the main educational outcomes analyses comprised 9545 children from the ALSPAC core sample who had complete education data.

**Results** Overall, of the 12 868 ALSPAC participants linked to NPD data, 137 had a CLA record and a further 209 a CIN record during adolescence. These children were more disadvantaged than their peers and had little active study participation beyond infancy. In the main educational outcomes analyses, achievement of 5+ good GCSEs was low in the CLA (OR 0.14, 95% CI 0.05 to 0.35) and CIN (0.11, 0.05 to 0.27) groups relative to their peers. Measured early life factors explained little of this difference.

**Conclusions** Data linkage enabled the study of educational outcomes in children with social services contact. These children had substantially worse educational outcomes relative to their peers, for reasons likely to be multifactorial.

## Strengths and limitations of this study

► We link a population-based birth cohort study (Avon Longitudinal Study of Parents and Children (ALSPAC)) to social care and educational records, and demonstrate that record linkage offers a means to identify vulnerable children in a cohort and increase their inclusion in research.

► The children in ALSPAC who had been looked-after (in public care) were broadly representative in terms of their care characteristics of children nationally of the same age who had been looked-after.

► We were only able to identify children who had been in care or in need during adolescence.

► Cohort data availability for children with social care records in adolescence was low beyond infancy.

prior to contact with services as opposed to later influences is unclear. Outcomes mainly resulting from early adversity may be less amenable to change through social care interventions, requiring alternative prevention strategies. These children are challenging to study using traditional research methods. A recent Children's Commissioner for England report highlights that vulnerable children are 'absent or poorly measured in national studies',[11] and children's social care is a difficult area in which to conduct randomised controlled trials.[12] Further, those who experience extreme adversity are likely under-represented in birth cohort studies due to low recruitment and high attrition, and identification of vulnerable children is challenging due to reliance on parental report.

Children with social services contact in England do, however, have high levels of administrative data. The term 'in need' refers to children who have been referred to and assessed by social services and found to be 'unlikely to achieve or maintain a reasonable

## INTRODUCTION

Children with social services contact, including those in public care, are at higher risk of poor outcomes than their peers, including low educational attainment, substance abuse and mental illness.[1–10] The extent to which this reflects early life adversity

level of health or development, or whose health and development is likely to be significantly or further impaired, without the provision of services; or a child who is disabled'.[13] Almost 390 000 children are currently classified as in need.[14] Some children in need may enter the public care system and become a 'looked-after' child. Presently over 72 000 children are looked-after,[15] with the majority placed with foster carers.[1]

While routine statistics using social care data can highlight poor outcomes, for example low average educational attainment, they lack information on early life and family characteristics.[1 16 17] These types of data are readily available in birth cohort studies. Linking cohort data to social care records could therefore provide a means of identifying children in need and looked-after without reliance on parental report. Further, using additional linked data to measure outcomes potentially enables the child's inclusion in analyses even if their family have stopped actively participating in the cohort study.

We use record linkage to a birth cohort to examine the effect of being in need or looked-after in adolescence on educational outcomes at age 16 years: the low attainment of many in need and looked-after children at this age is a concern as it can compound their disadvantaged childhoods to limit future education, employment and general life chances.[18]

## METHODS

### Data

#### Avon Longitudinal Study of Parents and Children

Pregnant women living in and around the city of Bristol, UK with expected date of delivery April 1991 to December 1992 were eligible to participate in the Avon Longitudinal Study of Parents and Children (ALSPAC). There were 14 541 pregnancies enrolled, resulting in 13 988 children alive at 1 year, including 13 972 singletons and twins. This 'core sample' was later bolstered by further eligible children: an additional 713 from age 7 to 18 years, and to date 183 since age 18 years. The mothers, their partners and the study children are studied via questionnaires and clinic visits. Teachers also completed questionnaires on the children. Further details are provided in cohort profiles[19 20] and searchable data dictionary.[21] For the main analyses on educational outcomes, the sample was restricted to: core, one child per family, with education data (n=9545, figure 1).

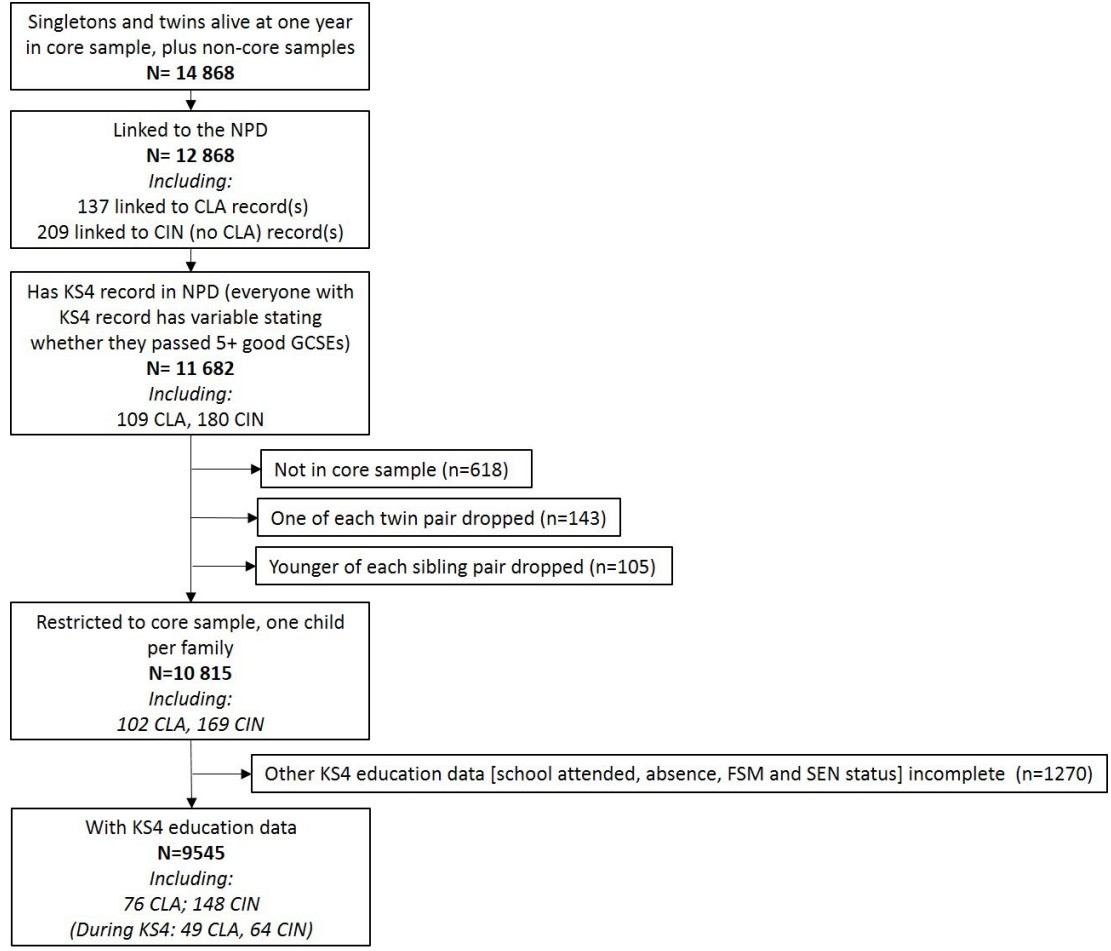

**Figure 1** Flowchart of sample. CIN, Children In Need; CLA, Children Looked-After; FSM, free school meals; GCSE, General Certificate of Education; KS4, Key Stage 4; NPD, National Pupil Database; SEN, special educational needs.

When study children reached age 18 years, they were sent 'fair processing' materials which described ALSPAC's intended use of their health and administrative records, and gave a clear means to object.[22] Education data were not extracted for participants who objected, or who were not sent fair processing materials.

### Linkage data

Data on children who are looked-after, or have been referred as a child in need, are collected annually via the Children Looked-After (CLA) Data Return[23] and the Children in Need (CIN) Census.[24] The CIN Census covers all children referred to children's social services even if no further action is taken. The CLA Return and the CIN Census have been linked to the National Pupil Database (NPD), a repository of education data for schools in England,[25] since their 2005/2006 and 2008/2009 data collections, respectively. ALSPAC has an established link to the NPD, and thus to any post-2005 CLA or post-2008 CIN record for participants in the NPD. Earlier CLA records were also obtained for those with a post-2005 record. However, CLA data collection was only on a random one-third sample of looked-after children from 1998 to 2003, meaning no records exist for many looked-after children in this period.[23] Insufficient identifiers exist within the CLA dataset to enable linkage of ALSPAC to pre-2005 CLA records for those without a post-2005 record.

We also obtained CLA records for all individuals in the CLA Return of a similar age (born January 1991 to December 1992) to form two comparison groups: (1) ever looked-after in England (n=43 938); (2) ever looked-after in the four local authorities that approximate the ALSPAC recruitment area (Bristol City; South Gloucestershire; North Somerset; Bath and North East Somerset) (n=713).

### Measures

#### Educational outcomes

Pupils in England study General Certificate of Education (GCSE) courses during Key Stage 4 (KS4) of their education (years 10 and 11, aged 14–16 years) and take GCSE exams at the end of year 11. The oldest ALSPAC children sat their GCSE exams in 2007, the youngest in 2009. Our main outcomes were two measures of attainment. First, a binary measure: achievement of 5+ good GCSEs (grades A*-C, including English and Maths). Second, a continuous measure: capped point score, expressed as a percentage of the maximum possible capped point score (based on the eight best grades obtained, with each grade assigned a numerical value).[26] Secondary educational outcomes included: persistent absence (≥10% of half days); special educational needs (SEN) status (see online supplementary text for definitions of the different SEN categories); and school mobility (whether child joined school during KS4).

#### Contact with children's social care services

Contact with children's social care services (referred to as 'social care status' hereafter) was summarised in two variables. The first specified whether a child had any post-2005 CLA record(s) or post-2008 CIN record(s) (ie, was looked-after or referred to social care services at any time for which we have linked social care data). The second summarised social care status during KS4 only. This restriction was necessary for the educational outcomes analyses to ensure our exposure preceded our outcome, plus these are the only two school years with CLA data coverage for all children in our sample (online supplementary table A). By definition children who are looked-after are also in need but we use in need to refer to children with a CIN but not a CLA record. The reference group comprised children with a KS4 record in the NPD who had no linked social care record.

Variables related to being in care or in need were derived from the linked data as follows. CIN Census: category of need; age referred. CLA Return: category of need; age first period of care (POC) started (POC is a period of time when child is continuously looked-after by the local authority); number of POC and episodes of care (a POC is comprised of 'episodes', each representing a period of being looked-after under the same legal status and in the same placement); placement type (foster; children's home/residential home/residential school; other (no further disaggregation possible due to small numbers)).

### Covariates

These included child age and sex, plus measures related to family socioeconomic position (SEP). Early life exposures included maternal age at delivery, and measures reported by the mother during pregnancy: highest educational qualification; financial difficulties; housing tenure; partner status; smoking; alcohol intake; social support; and depressive symptoms.[27] Later measures of SEP (during KS4) were obtained from the NPD: receipt of free school meals (FSM)[28]; and child's residential neighbourhood deprivation measured by the Income Deprivation Affecting Children Index.[29] More details in online supplementary text.

### Statistical analyses

Descriptive statistics were used to: summarise the social care data linked to ALSPAC children; compare the ALSPAC looked-after sample to the two non-ALSPAC looked-after comparison groups; compare child, maternal and SEP characteristics by social care status; describe questionnaire completion rates by social care status.

Associations between social care status and educational outcomes were examined using multilevel regression models (individual level 1, school level 2). Linear models were used for capped point score, logistic for attainment of 5+ good GCSEs. Associations were adjusted for age and sex (model 1), then also for KS4 measures (FSM, neighbourhood deprivation, school mobility) (model 2), or for early life exposures (model 3). We then adjusted for all KS4 and early life variables (model 4). Multiple imputation using chained equations was used to impute missing data (online supplementary table B) for the educational

outcomes analyses sample (n=9545). One hundred datasets were imputed.

In sensitivity analyses, models were restricted to children with no SEN (n=8145) or no disability (n=9506). Social care status at any time was also considered. Finally, we described associations between social care characteristics (eg, placement type, reason for being in need) and capped point score in those with CIN or CLA records: to maximise sample size, we included all those who had these records at any time and who had capped point score data.

### Patient and public involvement

Patients, the public and study participants were not directly involved in this study. Some ALSPAC participants are members of a committee which meets bimonthly to provide insights and advice on general ALSPAC study design, methodology and acceptability for participants.

### RESULTS
### Children in ALSPAC with social care records

Of those with a post-2008 CIN (but no CLA) record (n=209), the most common needs at referral were child disability, abuse or neglect, and family in acute stress. Of those with a post-2005 CLA record (n=137), the most common primary need was abuse or neglect (online supplementary tables C and D). Median total time in care was 2.6 years. Foster care was the most common placement type.

### Comparison to non-ALSPAC looked-after children

The ALSPAC children with CLA records were generally similar to those of children born at the same time who were ever in care in England (comparison group 1) or in the area in and around Bristol (comparison group 2) in terms of primary need (online supplementary table E). Importantly, many of those who had ever had a CLA record in the two comparison groups (36% of group 1; 42% of group 2) had left care before the age of 12 years (the youngest age at which we were able to link CLA records to ALSPAC).

### Availability of cohort data

Maternal questionnaire response rates were highest for participants with no social care record and lowest for those with a CLA record at all time points. Differences generally widened over time (figure 2). Patterns were similar for partner and child, but not teacher, questionnaires (online supplementary figure 1A-D).

### Educational outcomes at 16 years

Of the 9545 children in these analyses, 49 had CLA and 64 CIN (no CLA) records during KS4. These groups were more disadvantaged than their peers in early life and during KS4 (table 1). They were more likely to have joined their school recently.

Of those with CIN or CLA records, <15% passed 5+ good GCSEs compared with >50% of their peers. Mean percentage scores were also markedly lower (table 2). They were more likely to have SEN and persistent absence rates were higher, particularly for the in need group. Adjustment for school absence, neighbourhood deprivation, and receipt of FSM attenuated associations slightly for the CIN group but had less of an impact for the CLA group (table 3). Adjustment for early life maternal and SEP factors had more of an attenuating effect for the CLA than the CIN group. Attainment differences between these groups and their peers remained in the fully adjusted model.

In sensitivity analyses, when social care records at any time were considered, patterns were similar for the CLA group (n=76), while the CIN group (n=148) tended to do better than when restricted to only those who were in need during KS4 (table 2). When the sample excluded those with SEN or disability, results were similar to those of the main analyses (online supplementary tables F and G).

Estimates of the relationship between social care characteristics and attainment were imprecise due to small numbers. Those in foster placements had higher capped percentage scores (mean 37.0, 95% CI 30.7 to 43.2, n=64) than those in children's/residential homes/residential schools (28.3, 14.7 to 42.0, n=12). With regards need status, 'child disability' was associated with the lowest attainment for the CIN group and 'socially unacceptable behaviour' for the CLA group. For both CLA and CIN groups, those in the 'parental illness/disability' category had the highest attainment. However, CIs were wide and overlapping.

### DISCUSSION

Children who were looked-after or in need during KS4 had low attainment at age 16 years. The early life exposures we considered were not a major explanatory factor. We believe this is the first time linkage to the CLA Return and CIN Census has been used to identify birth cohort participants who were looked-after or in need during adolescence. As linkage data were also used for outcome measures, participants could be included even if their families no longer actively participated in the cohort study. Record linkage therefore allowed vulnerable children to not only be included in research but to be the focus of it. However, the identification and inclusion of in need and looked-after children in research using record linkage does have challenges.

For cohort studies in England with relevant permissions, linkage to the CLA Return and CIN Census via the NPD offers a convenient means of identifying participants who have been in need or looked-after. For cohorts younger than ALSPAC, this method would allow identification of social care records that cover most, if not all, of participants' childhoods. However, in ALSPAC, we were only able to link to records covering a period during adolescence. Consequently, outcomes at younger ages cannot be examined by social care status in ALSPAC using this

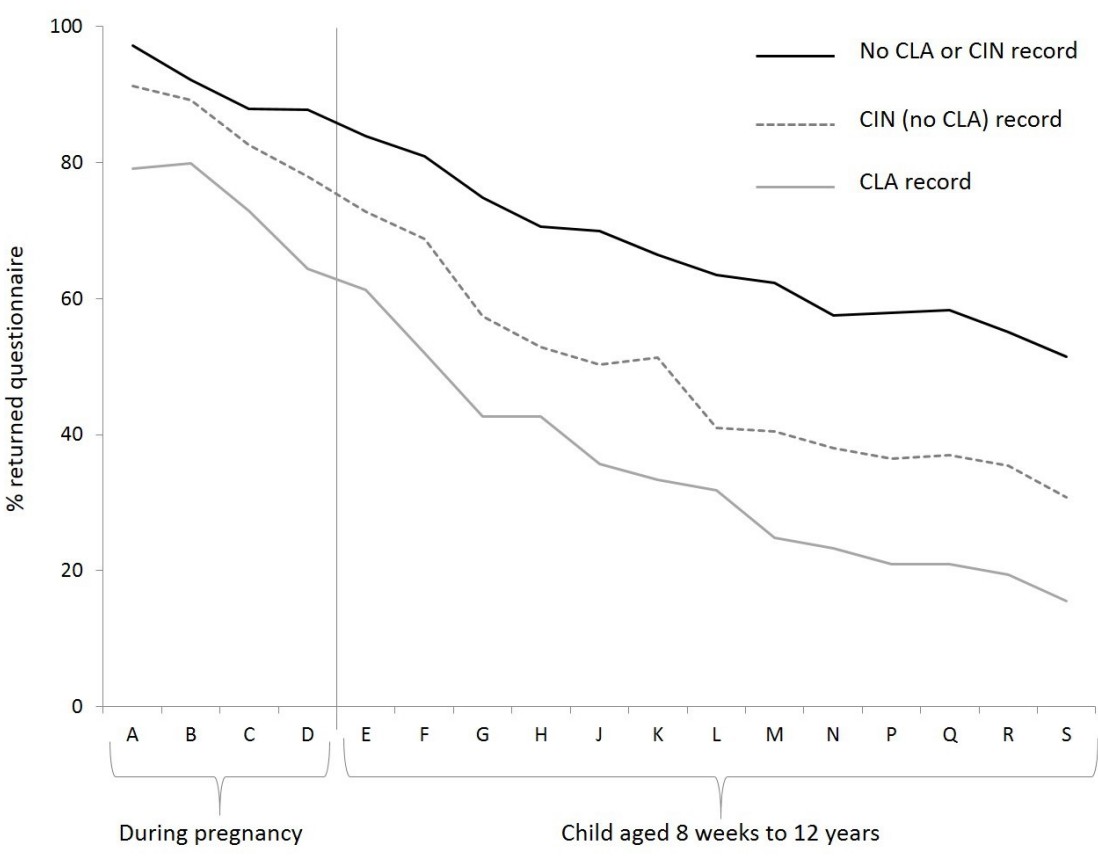

Notes on figure:

Sample restricted to mothers in the core sample, whose child has been linked to the National Pupil Database.

Mothers may not have completed every question within each questionnaire.

**Figure 2** Maternal questionnaire response rates by child social care status. CIN,Children In Need; CLA, Children Looked-After.

method. Of the looked-after children in England the same age as the ALSPAC participants, we found around 40% had left the care system by the age of 12 years. Consequently, our reference group likely includes children who were looked-after or in need at younger ages only.

Examination of questionnaire response rates showed the value of using linked outcome data to increase the inclusion of vulnerable children in research: there was little questionnaire data available beyond infancy for participants with social care records in adolescence. In this current study, we examined educational outcomes at age 16 years, obtained from the NPD. The association between social care status and other later outcomes available from linked data could also be investigated using ALSPAC, such as mental illness or entry into higher education.

ALSPAC participants with CLA and CIN records in adolescence had lower educational attainment than their peers in the reference group. In the most recent

national data available, attainment patterns by social care status broadly reflect these findings.[1] We found persistent absence rates to be considerably lower for those looked-after than those in need during KS4. Similarly, in the national data (on pupils of all ages) 9% of looked-after children were persistent absentees and 28% of children in need.[1] Therefore, although our participants were in KS4 around 10 years ago and the number with social care records small, the patterning of educational characteristics by care status is broadly similar to the present-day situation.

Using both ALSPAC questionnaire data and measures from the NPD, we found a persistence of disadvantage from early life to adolescence for participants with CIN and CLA records. Social disadvantage is known to be strongly associated with poorer educational attainment,[30][31] and our SEP measures were strongly related to the educational outcomes. Adjustment for them attenuated associations slightly but the low attainment of the

**Table 1** Summary of maternal, family and child characteristics, by social care status of child during Key Stage 4

| | Child's social care status during KS4 | | |
| --- | --- | --- | --- |
| | No CLA/CIN record n=9432 | CIN (no CLA) record n=64 | CLA record n=49 |
| | % (95% CI) | | |
| **Maternal and family characteristics during pregnancy*** | | | |
| Maternal age (at delivery) | | | |
| ≤23 years | 18.4 (17.7 to 19.2) | 39.1 (27.0 to 51.4) | 28.6 (15.4 to 41.7) |
| >33 years | 12.3 (11.6 to 12.9) | 7.8 (1.1 to 14.6) | 14.3 (4.1 to 24.5) |
| Relationship status | | | |
| Married | 75.0 (74.1 to 75.9) | 53.8 (40.8 to 66.7) | 49.5 (34.3 to 64.8) |
| Resident partner | 16.5 (15.7 to 17.3) | 17.0 (6.9 to 27.1) | 27.4 (13.5 to 41.3) |
| Non-resident/no partner | 8.5 (7.9 to 9.1) | 29.2 (17.3 to 41.2) | 23.1 (9.9 to 36.3) |
| Highest maternal education | | | |
| A level or degree | 30.7 (29.8 to 31.7) | 10.8 (2.3 to 19.3) | 11.1 (0.9 to 21.4) |
| O level | 36.5 (35.5 to 37.5) | 40.7 (27.2 to 54.1) | 26.3 (12.1 to 40.6) |
| Vocational/none | 32.8 (31.8 to 33.8) | 48.5 (34.6 to 62.5) | 62.5 (47.2 to 77.9) |
| Financial difficulties | | | |
| Highest quartile | 21.2 (20.3 to 22.1) | 42.2 (27.6 to 56.8) | 46.8 (30.4 to 63.2) |
| Housing tenure | | | |
| Owned/mortgaged | 73.7 (72.8 to 74.7) | 54.3 (41.4 to 67.3) | 33.7 (19.2 to 48.2) |
| Maternal smoking | | | |
| Yes | 26.6 (25.7 to 27.6) | 41.3 (28.2 to 54.3) | 58.6 (42.3 to 75.0) |
| Maternal alcohol—first trimester, ≥1 unit per week | | | |
| Yes | 15.2 (14.4 to 15.9) | 17.8 (7.6 to 28.0) | 21.7 (8.0 to 35.3) |
| Maternal alcohol—second trimester, ever ≥4 units in 1 day | | | |
| Yes | 16.9 (16.1 to 17.6) | 26.7 (14.8 to 38.5) | 21.1 (7.6 to 34.6) |
| Depression score | | | |
| Highest quartile | 23.4 (22.5 to 24.3) | 29.4 (16.4 to 42.4) | 47.8 (31.5 to 64.1) |
| Low social support | | | |
| Yes | 10.3 (9.6 to 11.0) | 20.8 (8.7 to 32.8) | 25.9 (10.6 to 41.2) |
| **Child, school and neighbourhood characteristics during KS4*** | | | |
| Sex | | | |
| Female | 49.6 (48.6 to 50.6) | 51.6 (39.0 to 64.2) | 49.0 (34.5 to 63.5) |
| Age at start of year 11 | | | |
| Mean (years) | 15.5 (15.4 to 15.5) | 15.5 (15.4 to 15.6) | 15.5 (15.4 to 15.5) |
| In receipt of free school meals | | | |
| Yes | 6.1 (5.6 to 6.6) | 26.6 (15.4 to 37.7) | 10.2 (1.4 to 19.0) |
| Joined school during KS4 | | | |
| Yes | 1.4 (1.1 to 1.6) | 7.8 (1.1 to 14.6) | 12.2 (2.7 to 21.8) |
| Neighbourhood deprivation (IDACI) | | | |
| Low, <10% | 43.9 (42.9 to 44.9) | 20.3 (10.2 to 30.4) | 28.6 (15.4 to 41.7) |
| High, ≥40% | 10.1 (9.5 to 10.7) | 25.0 (14.1 to 35.9) | 20.4 (8.7 to 32.1) |

*For brevity, not all categories are presented for each categorical variable.
CIN, Children In Need; CLA, Children Looked-After; IDACI, Income Deprivation Affecting Children Index; KS4, Key Stage 4.

CLA and CIN groups remained. We are not considering the SEP measures as confounders but rather part of the complex causal pathway from early life adversity through to poor educational attainment. It is notable that many of the mothers of the children with social care records had very low educational attainment themselves.

**Table 2** Educational attainment, persistent absence and special educational needs by child social care status

| | Social care status during KS4 | | | Social care status any time | | |
|---|---|---|---|---|---|---|
| | No CLA/CIN record n=9432 | CIN (no CLA) record n=64 | CLA record n=49 | No CLA/CIN record n=9321 | CIN (no CLA) record n=148 | CLA record n=76 |
| Educational attainment | % or mean (95% CI) | | | % or mean (95% CI) | | |
| 5+A*-C GCSEs including English and Maths | 53.0 (52.0 to 54.0) | 10.9 (3.1 to 18.8) | 12.2 (2.7 to 21.8) | 53.3 (52.3 to 54.4) | 19.6 (13.1 to 26.1) | 10.5 (3.5 to 17.6) |
| Capped percentage point score | 68.9 (68.5 to 69.3) | 37.4 (31.3 to 43.5) | 34.9 (27.4 to 42.3) | 69.1 (68.7 to 69.5) | 47.7 (43.8 to 51.7) | 33.9 (28.2 to 39.6) |
| Special educational needs (SEN)* | | | | | | |
| School action | 8.5 (7.9 to 9.1) | 12.5 (4.2 to 20.8) | n<5 | 8.4 (7.8 to 9.0) | 16.2 (10.2 to 22.2) | 9.2 (2.6 to 15.9) |
| School action plus | 3.1 (2.8 to 3.5) | 15.6 (6.5 to 24.8) | 24.5 (12.0 to 37.0) | 3.1 (2.7 to 3.4) | 8.1 (3.7 to 12.6) | 21.1 (11.7 to 30.4) |
| Statement of special educational needs | 2.4 (2.1 to 2.7) | 46.9 (34.3 to 59.4) | 24.5 (12.0 to 37.0) | 2.3 (2.0 to 2.6) | 22.3 (15.5 to 29.1) | 35.5 (24.5 to 46.5) |
| Persistent absence | 6.8 (6.3 to 7.3) | 32.8 (21.0 to 44.6) | 18.4 (7.1 to 29.6) | 6.7 (6.2 to 7.2) | 19.6 (13.1 to 26.1) | 21.1 (11.7 to 30.4) |

*For definitions of these SEN categories, please see online supplementary text.
CIN, Children In Need; CLA, Children Looked-After; GCSE, General Certificate of Education; KS4, Key Stage 4.

**Table 3** Association between child social care status and educational outcomes, with adjustment for early life and KS4 variables

| Attainment Outcome | Care status during KS4 | Model 1* (age and sex) | Model 2† (KS4 variables) | Model 3‡ (early life variables) | Model 4§ (fully adjusted) |
|---|---|---|---|---|---|
| 5+A*-C GCSEs including English and Maths | | OR (95% CI) | OR (95% CI) | OR (95% CI) | OR (95% CI) |
| | Not CIN or CLA | Ref | Ref | Ref | Ref |
| | CIN (not CLA) | 0.11 (0.05 to 0.27) | 0.17 (0.07 to 0.40) | 0.15 (0.06 to 0.36) | 0.19 (0.08 to 0.46) |
| | CLA | 0.14 (0.05 to 0.35) | 0.14 (0.06 to 0.36) | 0.25 (0.10 to 0.63) | 0.24 (0.09 to 0.63) |
| Capped percentage point score | | Coeff (95% CI) | Coeff (95% CI) | Coeff (95% CI) | Coeff (95% CI) |
| | Not CIN or CLA | Ref | Ref | Ref | Ref |
| | CIN (not CLA) | −22.1 (−26.7 to −17.5) | −14.1 (−18.4 to −9.8) | −18.4 (−22.6 to −14.1) | −13.1 (−17.1 to −9.0) |
| | CLA | −28.4 (−33.5 to −23.3) | −25.0 (−29.7 to −20.3) | −21.9 (−26.6 to −17.2) | −20.6 (−25.0 to −16.1) |

*Adjusted for child age and sex.
†Adjusted for child age and sex, plus KS4 time-point variables (persistent school absence, in receipt of free school meals, school mobility, IDACI of residential neighbourhood).
‡Adjusted for child age and sex, plus early life (maternal age at delivery, education, partner status, housing tenure, financial difficulties, smoking, alcohol, depression, social support).
§Adjusted for child age and sex, plus KS4 and early life variables.
CIN, Children In Need; CLA, Children Looked-After; IDACI, Income Deprivation Affecting Children Index; KS4, Key Stage 4; SEP, socioeconomic position.

Alcohol and tobacco, the most commonly used substances in pregnancy, can cross the placenta and alter normal brain development.[32] In our sample, those with social care records had higher levels of exposure to these substances than their peers. Those with CIN records were the most likely to have ever been exposed to ≥4 units of alcohol in 1 day. Exposure to this level of alcohol has previously been found to be negatively associated with educational attainment in the ALSPAC sample.[33 34] However, in our analyses, adjustment for maternal alcohol use did not alter the associations observed between social care status and educational attainment. This may be due in part to our binary alcohol measures (necessary due to small numbers) failing to accurately capture exposure, and not identifying those at highest risk. This is an important limitation as many children in the care system have foetal alcohol syndrome, a condition which is often undiagnosed and is the most common, non-genetic cause of learning disability in the UK.[35 36] The majority of participants with CLA records had a mother who smoked during pregnancy, and this exposure was negatively associated with attainment. However, there is debate as to whether maternal smoking during pregnancy is a direct cause of poorer child educational attainment, or is instead a strong marker of socioeconomic disadvantage.[37–39]

Overall, little of the poor educational outcomes in the looked-after and in need groups appeared to be explained by the early life exposures we considered. This could suggest there is scope for later experiences, including social care, to improve outcomes. However, other early life exposures, or genetic factors, that we have not considered could be of importance.

While aspects of care itself could be important contributors to educational outcomes, ascertaining direction of causality in the relationship between child behaviours, care characteristics and educational outcomes is difficult. As expected, we found children in foster care had higher attainment than those not in family based care: the latter children are likely to be those whose foster placements have broken down, reflecting complex additional needs and challenging behaviours. Further, foster carers may have greater commitment and longer-term interest in the child than group care staff.[40] We were unable to consider placement stability, which has previously been shown to be beneficial.[41] However, in concordance with previous studies, school mobility was associated with lower attainment[42 43] and children with CLA or CIN records were much more likely to have changed school during KS4 than their peers.

The relatively high proportion of looked-after and in need children with SEN or disability did not appear to explain the low average attainment of these groups. Similarly, in the national data, looked-after children with no identified SEN made less educational progress than non-looked-after children.[1] It is important to note that the attainment gap between looked-after and in need children and their peers is apparent from a young age, often before the child enters care.[1 41] Being looked-after

may not be the principal cause of poor attainment, rather it is a marker of extreme childhood adversity, which is itself associated with poor outcomes. For children who have experienced adversity in childhood, being in care is often beneficial for their education.[17 18 41]

Strengths of this study include the use of a novel method to identify vulnerable adolescents in a population-based cohort, and objective outcome measures. Limitations include incomplete ascertainment of social care record status, little cohort data beyond early childhood for those with social care records, and small numbers. Children who experience the most disadvantaged starts in life are likely under-represented in ALSPAC as their mothers would have been least likely to attend antenatal appointments, which is where many mothers were recruited to the study.

## CONCLUSIONS

Data linkage provides a means of identifying children with social services contact in cohort studies and of increasing their inclusion in research. The poor educational outcomes of the ALSPAC adolescents with social care records did not appear to be substantially explained by the early life exposures we considered. Further research, ideally with social care data across the lifecourse, would help identify which factors are important in explaining the poor educational attainment of these vulnerable children, and would help inform the development of effective interventions.

**Acknowledgements** We are extremely grateful to all the families who took part in this study, the midwives for their help in recruiting them, and the whole ALSPAC team, which includes interviewers, computer and laboratory technicians, clerical workers, research scientists, volunteers, managers, receptionists and nurses.

**Contributors** JM and AB conceived the study. AT and JM developed the research question. AT conducted the analyses, interpreted the data and drafted the manuscript. JM and DW helped interpret the data and critically revised the paper. AB critically revised the paper.

**Funding** Core support for ALSPAC is provided by the UK Medical Research Council and Wellcome (Grant reference: 102215/2/13/2) and the University of Bristol. The Wellcome Trust (WT086118) funded ALSPACs linkage infrastructure through the Project to Enhance ALSPAC through Record Linkage (PEARL). This work was supported by the Elizabeth Blackwell Institute for Health Research, University of Bristol and the Wellcome Trust Institutional Strategic Support Fund (Grant number: 105612/Z/14/Z).

**Competing interests** None declared.

**Patient consent for publication** Not required.

**Ethics approval** Ethical approval for ALSPAC was obtained from the ALSPAC Ethics and Law Committee and Local Research Ethics Committees (www.bristol.ac.uk/alspac/researchers/research-ethics/).

**Provenance and peer review** Not commissioned; externally peer reviewed.

**Data availability statement** Access to ALSPAC data is through a system of managed open access (http://www.bristol.ac.uk/alspac/researchers/access/).

**ORCID iD**

Alison Teyhan http://orcid.org/0000-0002-4965-3139

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
