## [Reviewer comments · BMJ Open]

ARTICLE DETAILS

TITLE (PROVISIONAL)	Early-life adversity, contact with children's social care services, and educational outcomes at age 16 years: UK birth cohort study with linkage to national administrative records
AUTHORS	Teyhan, Alison; Boyd, Andy; Wijedasa, Dinithi; Macleod, John

VERSION 1 – REVIEW

REVIEWER	Philip Wilson University of Aberdeen Scotland
REVIEW RETURNED	22-Mar-2019

GENERAL COMMENTS	This paper is a worthy attempt to describe and quantify some of the factors underlying the poor educational attainment of looked-after children and those identified as being in need. The authors clearly describe their methodology and some of the difficulties inherent in a data linkage study involving a cohort in which the administrative data from early life is very incomplete. The findings are in general unsurprising, but this should not detract from the methodological value of the work. My guess is that the methods will be a useful guide to similar studies with more contemporary cohorts such as Millennium and Growing Up in Scotland. More complete population coverage might be obtained using some of the Scandinavian registers but some of the early life exposures would not be available with these resources. My main concern about the paper is the repeated statement at several points in the manuscript that early life exposures did not explain much of the association between looked after status and poor outcomes. The authors used a very restricted number of antenatal early life variables available from the ALSPAC dataset in their analyses, and I am particularly concerned about the failure to include antenatal alcohol and medication use. A number of previous ALSPAC publications have utilised these data. A high proportion of looked after children have fetal alcohol syndrome (or fetal alcohol spectrum disorder), although this diagnosis was often not recorded, particularly among children born in the early 90s - and the diagnosis in children without dysmorphic features is very difficult. Other antenatal adversities and genetic factors are completely ignored in the discussion and their inclusion would, in my view, greatly add to the strength of the work. Antenatal smoking rates were substantially higher in the LAC and CIN groups, and, while the association with poor outcomes may relate to confounding (see the e-risk study of Maughan et al 2005) it is still noteworthy, as is the low educational attainment of the mothers of these children. These are important observations and should be discussed.
--

	Minor point: Page 6, line 42 I think the dates should be 1991 and 1992.
REVIEWER	Marc Winokur Colorado State University, United States
REVIEW RETURNED	09-Apr-2019
GENERAL COMMENTS	The author's primary contribution is the linking of a population-based birth cohort study to social care and educational records. The author's rightly highlight that record linkage offers a means to identify vulnerable children in a cohort and increase their inclusion in research. Furthermore, it is essential to link data across systems given the increase in multi-system involved youth. The other contribution is the focus on educational achievement of youth in public care, which is an under-studied outcome with major implications for the well-being of these youth. The limitations are aptly described regarding the low availability of cohort data beyond infancy for children with social care records in adolescence. Although the main findings that children looked after and children in need have lower educational attainment than their peers is not surprising, it is important nonetheless to document and provide a magnitude of the effect. This information will allow schools and child welfare agencies to work together to develop interventions to address this inequity.

VERSION 1 – AUTHOR RESPONSE

Reviewer: 1

- This paper is a worthy attempt to describe and quantify some of the factors underlying the poor educational attainment of looked-after children and those identified as being in need. The authors clearly describe their methodology and some of the difficulties inherent in a data linkage study involving a cohort in which the administrative data from early life is very incomplete. The findings are in general unsurprising, but this should not detract from the methodological value of the work. My guess is that the methods will be a useful guide to similar studies with more contemporary cohorts such as Millennium and Growing Up in Scotland. More complete population coverage might be obtained using some of the Scandinavian registers but some of the early life exposures would not be available with these resources.

Thank you for your positive comments. We recognise that there are limitations to our work but we believe it has value, not least in highlighting to other cohort studies the potential to increase the inclusion of vulnerable children in research through record linkage.

- My main concern about the paper is the repeated statement at several points in the manuscript that early life exposures did not explain much of the association between looked after status and poor outcomes. The authors used a very restricted number of antenatal early life variables available from the ALSPAC dataset in their analyses.

On relection, we agree with the reviewer that these statements were misleading. We have altered the text to emphasise that we only adjust for some early-life exposures. For example, in the first paragraph of the discussion, we now say:

“The early-life exposures we considered were not a major explanatory factor.”

And we have re-focused the conclusion on the need for future research in this area:

“The poor educational outcomes of the ALSPAC adolescents with social care records did not appear to be substantially explained by the early-life exposures we considered. Further research, ideally with social care data across the lifecourse, would help identify which factors are important in explaining the poor educational attainment of these vulnerable children, and would help inform the development of effective interventions.”

- I am particularly concerned about the failure to include antenatal alcohol and medication use. A number of previous ALSPAC publications have utilised these data. A high proportion of looked after children have fetal alcohol syndrome (or fetal alcohol spectrum disorder), although this diagnosis was often not recorded, particularly among children born in the early 90s - and the diagnosis in children without dysmorphic features is very difficult. Other antenatal adversities and genetic factors are completely ignored in the discussion and their inclusion would, in my view, greatly add to the strength of the work.

Thank you for pointing out the importance of including measures of maternal alcohol use in pregnancy in our analyses. We have selected two measures to include in our models (both of which have been used in previous ALSPAC studies of child educational attainment): frequency of consumption of alcohol in first trimester; and consumption of ≥ 4 units of alcohol in one day during second trimester. Descriptive summaries of these variables have been added to Table 1 (showing, for example, that those in the CIN group were the most likely to have a mother who drank ≥ 4 units per day in the second trimester) and Supplementary Table B (shows overall summary, not by social care status). Models 3 and 4 now also include both of these alcohol variables (results shown in Table 3, and Supplementary Tables F and G). Adjustment for these additional variables did not alter the associations previously observed.

We have added a paragraph to the discussion which highlights that the children with social care records were more likely to have been exposed in utero to alcohol and tobacco:

“Alcohol and tobacco, the most commonly used substances in pregnancy, can cross the placenta and alter normal brain development (32). In our sample, those with social care records had higher levels of exposure to these substances than their peers. Those with CIN records were the most likely to have ever been exposed to ≥ 4 units of alcohol in one day. Exposure to this level of alcohol has previously been found to be negatively associated with educational attainment in the ALSPAC sample (33, 34). However, in our analyses, adjustment for maternal alcohol use did not alter the associations observed between social care status and educational attainment. This may be due in part to our binary alcohol measures (necessary due to small numbers) failing to accurately capture exposure, and not identifying those at highest risk. This is an important limitation as many children in the care system have foetal alcohol syndrome, a condition which is often undiagnosed and is the most common, non-genetic cause of learning disability in the UK (35, 36). The majority of participants with CLA records had a mother who smoked during pregnancy, and this exposure was negatively associated with attainment. However there is debate as to whether maternal smoking during pregnancy is a direct cause of poorer child educational attainment, or is instead a strong marker of socio-economic disadvantage (37-39).”

We agree that medication use in pregnancy could potentially differ by child social care status and be associated with child educational attainment. However, we do not feel we are able to include measures of medication use in the analyses for this paper. The vast majority (92%) of the ALSPAC mothers used medication during pregnancy (<https://www.ncbi.nlm.nih.gov/pubmed/15168103>).

Different medications have different biological mechanisms and their uses could be socially patterned in different ways. Due to the small numbers in our CLA and CIN exposure categories, the number of mothers using any one medication would be small and we do not have the power to unpick the effects of one medication from another. We also haven't considered exposure to illegal substances. Further, our sample size is too small to consider genetic effects which could also be an important factor. We instead added a paragraph to our discussion to acknowledge that there are many other early life factors that we have not been able to consider:

"Overall, little of the poor educational outcomes in the looked-after and in need groups appeared to be explained by the early-life exposures we considered. This could suggest there is scope for later experiences, including social care, to improve outcomes. However, other early-life exposures, or genetic factors, that we have not considered could be of importance."

- Antenatal smoking rates were substantially higher in the LAC and CIN groups, and, while the association with poor outcomes may relate to confounding (see the e-risk study of Maughan et al 2005) it is still noteworthy, as is the low educational attainment of the mothers of these children. These are important observations and should be discussed.

We have now included a discussion of the exposure to smoking in pregnancy results in our discussion (please see paragraph above, which also mentions the alcohol results). We have also highlighted the low educational attainment of the mothers of children with social care records:

"It is notable that many of the mothers of the children with social care records had very low educational attainment themselves."

- Minor point: Page 6, line 42 I think the dates should be 1991 and 1992.

Thank you for spotting this! We have corrected the dates.

Reviewer: 2

The author's primary contribution is the linking of a population-based birth cohort study to social care and educational records. The author's rightly highlight that record linkage offers a means to identify vulnerable children in a cohort and increase their inclusion in research. Furthermore, it is essential to link data across systems given the increase in multi-system involved youth. The other contribution is the focus on educational achievement of youth in public care, which is an under-studied outcome with major implications for the well-being of these youth. The limitations are aptly described regarding the low availability of cohort data beyond infancy for children with social care records in adolescence. Although the main findings that children looked after and children in need have lower educational attainment than their peers is not surprising, it is important nonetheless to document and provide a magnitude of the effect. This information will allow schools and child welfare agencies to work together to develop interventions to address this inequity.

Thank you for your positive comments.

VERSION 2 – REVIEW

REVIEWER	Philip Wilson University of Aberdeen, UK
REVIEW RETURNED	02-Sep-2019

GENERAL COMMENTS	I am happy with this amended manuscript. The authors have dealt with the initial concerns well.
---

REVIEWER	Marc Winokur Colorado State University, USA
REVIEW RETURNED	18-Aug-2019

GENERAL COMMENTS	Thank you for your careful consideration of the reviewer suggestions.
---